# The Waymo Open Sim Agents Challenge

**Nico Montali**     **John Lambert**     **Paul Mougin**     **Alex Kuefler**     **Nicholas Rhinehart**

**Michelle Li**     **Cole Gulino**     **Tristan Emrich**     **Zoey Yang**     **Shimon Whiteson**

**Brandyn White**          **Dragomir Anguelov**

Waymo LLC

## Abstract

Simulation with realistic, interactive agents represents a key task for autonomous vehicle software development. In this work, we introduce the Waymo Open Sim Agents Challenge (WOSAC). WOSAC is the first public challenge to tackle this task and propose corresponding metrics. The goal of the challenge is to stimulate the design of realistic simulators that can be used to evaluate and train a behavior model for autonomous driving. We outline our evaluation methodology, present results for a number of different baseline simulation agent methods, and analyze several submissions to the 2023 competition which ran from March 16, 2023 to May 23, 2023. The WOSAC evaluation server remains open for submissions and we discuss open problems for the task.

## 1   Introduction

Simulation environments allow cheap and fast evaluation of autonomous driving behavior systems, while also reducing the need to deploy potentially risky software releases to physical systems. While generation of synthetic sensor data was an early goal [19, 43] of simulation, use cases have evolved as perception systems have matured. Today, one of the most promising use cases for simulation is system safety validation via statistical model checking [1, 16] with Monte Carlo trials involving realistically modeled traffic participants, i.e., *simulation agents*.

Simulation agents are controlled objects that perform realistic behaviors in a virtual world. In this challenge, in order to reduce the computational burden and complexity of simulation, we focus on simulating agent behavior as captured by the outputs of a perception system, e.g., mid-level object representations [2, 79] such as object trajectories, rather than simulating the underlying sensor data [13, 37, 62, 76] (see Figure 1).

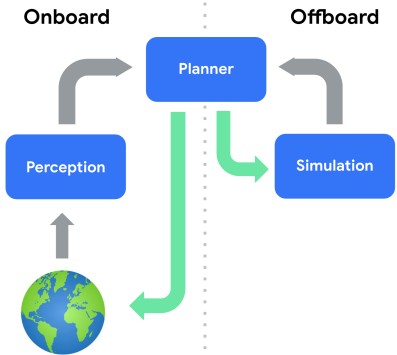

Figure 1: WOSAC models the simulation problem as simulation of mid-level object representations, rather than as sensor simulation.

A requirement for modeling realistic behavior in simulation is the ability for sim agents to respond to arbitrary behavior of the autonomous vehicle (AV). "Pose divergence" or "simulation drift" [3] is defined as the deviation between the AV's behavior in driving logs and its behavior during simulation, which may be represented through differing position, heading, speed, acceleration, and more. Directly replaying logged behavior of all other objects in the scene [32, 34, 35]

37th Conference on Neural Information Processing Systems (NeurIPS 2023) Track on Datasets and Benchmarks.

Table 1: A comparison of three autonomous-vehicle behavior related tasks which involve generation of a desired future sequence of physical states: trajectory forecasting, planning, and simulation. Note that observations $o_t \in \mathcal{O}$ include simulated agent and environment properties.

| Task | Multiple Object Categories | Outputs | Vehicle Kinematic Constraints | System Evaluation | System Objectives |
|---|---|---|---|---|---|
| Multi-Agent Trajectory Forecasting | ✓ | $\left((x_t, y_t, \theta_t, v_t^x, v_t^y)\right)_{t=1}^{T}$ | ✗ | Open-Loop | Kinematic accuracy and mode covering |
| AV Motion Planning | ✗ | $\left((x_t, y_t, \theta_t)\right)_{t=1}^{T}$ or controls | ✓ | Closed-Loop | Safety, comfort, progress |
| Agent and Environment Simulation | ✓ | $\left(o_t\right)_{t=1}^{T}; o_t \in \mathcal{O}$ | ✗ | Closed-Loop | Distributional realism |

under arbitrary AV planning may have limited realism because of this pose divergence. Such log-playback agents tend to heavily overestimate the aggressiveness of real actors, as they are unwilling to deviate from their planned route under any circumstances. On the other hand, rule-based agents that follow heuristics such as the Intelligent Driver Model (IDM) [65] are overly accommodating and reactive. We seek to evaluate and encourage the development of sim agents that lie in the middle ground, adhering to a definition of *realism* that implies matching the full distribution of human behavior.

To the best of our knowledge, to date there is no existing benchmark for evaluation of simulation agents. Benchmarks have spurred notable innovation in other areas related to autonomous driving research, especially for perception [6, 10, 24, 58], motion forecasting [6, 10, 22, 72, 78], and motion planning [19]. We believe a standardized benchmark can likewise spur dramatic improvements for simulation agent development. Among these benchmarks, those focused on motion forecasting are perhaps most similar to simulation, but all involve open-loop evaluation, which is clearly deficient compared to our closed-loop evaluation. Furthermore, we introduce realism metrics which are suitable to evaluating long-term futures. Relevant datasets such as the Waymo Open Motion Dataset (WOMD) [22] exist today that contain real-world agent behavior examples, and we build on top of WOMD to build WOSAC. In this challenge, we focus on a subset of the possible perception outputs, e.g., traffic light states or vehicle attributes are not modeled, but we leave this for future work.

The challenges our benchmark raises are unique, and if we can make real progress on it, we can show that we've solved one of the hard problems in self-driving. We have a number of open questions: Are there benefits to scene-centric, rather than agent-centric, simulation methods? What is the most useful generative modeling framework for the task? What degree of motion planning is needed for agent policies, and how far can marginal motion prediction take us? How can simulation methods be made more efficient? How can we design a benchmark and enforce various simulator properties? During our first iteration of the WOSAC challenge, user submissions have helped us answer a subset of these questions; for example, we observed that most methods found it most expedient to build upon state-of-the-art marginal motion prediction methods, i.e. operating in an agent-centric manner.

In this work, we describe in detail the Waymo Open Sim Agents Challenge (WOSAC) with the goal of stimulating interest in traffic simulation and world modeling. Our contributions are as follows:

- An evaluation framework for autoregressive traffic agents based on the approximate negative log likelihood they assign to logged data.

- An evaluation platform, an online leaderboard, available for submission at `https://waymo.com/open/challenges/2023/sim-agents/`.

- An empirical evaluation and analysis of various baseline methods, as well as several external submissions.

## 2 Related Work

**Multi-Agent Traffic Simulation** Simulators have been used to train and evaluate autonomous driving planners for several decades, dating back to ALVINN [43]. While simulators such as CARLA [19], SUMO [33], and Flow [73] provide only a heuristic driving policy for sim agents, they have still enabled progress in the AV motion planning domain [11, 12, 15]. Other recent simulators such as Nocturne [68] use a simplified world representation that consists of a fixed roadgraph and moving agent boxes.

Table 2: Existing evaluation methods for simulation agents. Entries are ordered chronologically by Arxiv timestamp. There is limited consensus in the literature regarding how multi-agent simulation should be evaluated.

| Evaluation Protocol | ADE or minADE | Offroad Rate | Collision Rate | Instance-Level Distribution Matching | Dataset-Level Distribution Matching | Spatial Coverage or Diversity | Goal progress or Completion |
|---|---|---|---|---|---|---|---|
| ConvSocialPool [17] | ✓ | | | ✓ | | | |
| Trajectron [29] | ✓ | | | ✓ | | | |
| PRECOG [48] | ✓ | | | ✓ | | | |
| BARK [4] | | ✓ | ✓ | | | | ✓ |
| SMARTS [82] | | | ✓ | | | | ✓ |
| TrafficSim [60] | ✓ | ✓ | ✓ | | | ✓ | |
| SimNet [3] | ✓ | | ✓ | | | | |
| Symphony [28] | ✓ | ✓ | ✓ | | ✓ | | |
| Nocturne [68] | ✓ | | ✓ | | | | ✓ |
| BITS [74] | | ✓ | ✓ | | ✓ | ✓ | |
| InterSim [59] | ✓ | | ✓ | | | | ✓ |
| MetaDrive [34] | | ✓ | ✓ | | | | ✓ |
| TrafficBots [79] | ✓ | | ✓ | ✓ | | | |
| WOSAC (Ours) | | ✓ | ✓ | ✓ | | | |

Simulation agent modeling is closely related to the problem of trajectory forecasting, as a sim agent could execute a set of trajectory predictions as its plan [4, 59]. However, as trajectory prediction methods are traditionally trained in open-loop, they have limited capability to recover from out of domain predictions encountered during closed-loop simulation [51]. In addition, few forecasting methods produce consistent joint future samples at the scene level [36, 48]. Sim agent modeling is also related to planning, as each sim agent could execute a replica of a planner independently [4]. However, each of these three tasks differ dramatically in objectives, outputs, and constraints (see Table 1).

**Learned Sim Agents** Learned sim agents in the literature differ widely in assumptions around policy coordination, dynamics model constraints, observability, as well as input modalities. While coordinated scene-centric agent behavior is studied in the open-loop motion forecasting domain [8, 9, 57], to the best of our knowledge, TrafficSim [60] is the only closed-loop, learned sim agent work to use a joint, scene-centric actor policy; all others operate in a decentralized manner without coordination [5, 28, 74], i.e., each agent in the scene is independently controlled by replicas of the same model using agent-centric inference. BITS and TrafficBots [74, 79] use a unicycle dynamics model and Nocturne uses a bicycle dynamics model [68] whereas most others do not specify any such constraint; others enforce partial observability constraints, such as Nocturne [68]. Other methods differ in the type of input, whether rasterized [3, 74] or provided in a vector format [28, 79]. Some works focus specifically on generating challenging scenarios [47], and others aim for user-based controllability [81]. Some are trained via pure imitation learning [3], while others include closed-loop adversarial losses [28, 60], or multi-agent RL [4, 34, 82] in order to learn to recover from its mistakes [51]. Some works such as InterSim [4, 28, 59, 74, 79, 82] use a goal-conditioned problem formulation, while others do not [3].

**Evaluation of Generative Models** Distribution matching has become a common way to evaluate generative models [18, 20, 27, 30, 31, 45, 46, 49, 52, 77], through the Fréchet Inception Distance (FID) [26]. Previous evaluation methods such as the Inception Score (IS) [53] reason over the entropy of conditional and unconditional distributions, but are not applicable in our case due to the multi-modality of the simulation problem. The FID improves the Inception Score by using statistics of real world samples, measuring the difference between the generated distributions and a data distribution (in the simulation domain, the logged distribution). However, FID has limited sensitivity per example due to aggregation of statistics over entire test datasets into a single mean and covariance.

**Evaluating Multi-Agent Simulation** There is limited consensus in the literature regarding how multi-agent simulation should be evaluated (see Table 2), and no mainstream existing benchmark exists. Given the importance of safety, almost all existing sim agent works measure some form of collision rate [3, 28, 59, 60, 68, 74], and some multi-object joint trajectory forecasting methods also measure it via trajectory overlap [36]. However, collision rate can be artificially driven to zero by static policies, and thus cannot measure realism. Quantitative evaluation of realism requires comparison with logged data. Such evaluation methods vary widely, from distribution matching of vehicle dynamics [28, 74], to comparison of offroad rates [28, 60, 74], spatial coverage and diversity [60, 74], and progress to goal [59, 68]. However, as goals are not observable, they are thus difficult to extract reliably. Requiring direct reconstruction of logged data through metrics such as Average Displacement Error (ADE) has also been proposed [3, 68], but has limited effectiveness because

there are generally multiple realistic actions the AV or sim agents could take at any given moment. To overcome this limitation, one option is to allow the user to provide multiple possible trajectories per sim agent, such as TrafficSim, which uses a minimum average displacement error (minADE) over 15 simulations. [60].

Recently, generative-model based evaluation has become more popular in the simulation domain, primarily through distribution matching metrics. Symphony [28] uses Jensen-Shannon distances over trajectory curvature. NeuralNDE [75] compares distributions of vehicle speed and inter-vehicle distance, whereas BITS [74] utilizes Wasserstein distances on agent scene occupancy using multiple rollouts per scene, along with Wasserstein distances between simulated and logged speed and jerk – two kinematic features which can encapsulate passenger comfort. The latter are computed as a distribution-to-distribution comparison on a dataset level, however, this type of metric has shown limited sensitivity in our experiments.

**Likelihood metrics** An alternative distribution matching framework is to measure point-to-distribution distances. [17] and [29] introduce a metric defined as the average negative log likelihood (NLL) of the ground truth trajectory, as determined by a kernel density estimate (KDE) [42, 50] over output samples at the same prediction timestep. This metric has found some adoption [54, 64], and we primarily build off of this metric in our work. We note that likelihood-based generative models of simulation such as PRECOG [48] and MFP [63] directly produce likelihoods, meaning that the use of a KDE on sampled trajectories to estimate likelihoods is not needed for such model classes. Concurrent work [79] also measures the NLL of the GT scene under 6 rollouts.

## 3 Traffic Simulation as Conditional Generative Modeling

Our goal is to encourage the design of traffic simulators by defining a data-driven evaluation framework and instantiating it with publicly accessible data. We focus on simulating agent behavior in a setting in which an offboard perception system is treated as fixed and given.

**Problem formulation.** We formulate driving as a Hidden Markov Model $\mathcal{H} = \big(\mathcal{S}, \mathcal{O}, p(o_t|s_t), p(s_t|s_{t-1})\big)$, where $\mathcal{S}$ denotes the set of unobservable true world states, $\mathcal{O}$ denotes the set of observations, $p(o_t|s_t)$ denotes the sampleable emission distribution, and $p(s_t|s_{t-1})$ denotes the hidden Markovian state dynamics: the probability of the hidden state transitioning from $s_{t-1}$ at timestep $t-1$ to $s_t$ at time $t$. Each $o_t \in O$ can be partitioned into AV- and environment-centric components that vary in time: $o_t = [o_t^{\mathrm{AV}}, o_t^{\mathrm{env}}]$. $\mathcal{O}_t^{\mathrm{env}}$ can in general contain a rich set of features, but for the purpose of our challenge, it contains solely the poses of the non-AV agents. We denote the true observation dynamics as $p^{\mathrm{world}}(o_t|s_{t-1}) \doteq \mathbb{E}_{p(s_t|s_{t-1})} p(o_t|s_t)$.

**The task.** The task to build a "world model" $q^{\mathrm{world}}(o_t|o_{<t}^c)$ of $p^{\mathrm{world}}(o_t|s_{t-1})$, $o_{<t}^c \doteq [o^{\mathrm{map}}, o^{\mathrm{signals}}, o_{-H-1}, \ldots, o_{t-1}]$, i.e., it denotes a context of a static map observation, traffic signal observations, and the observation history, with history length $H$.

**Task constraints:**

1. $q^{\mathrm{world}}$ must be autoregressive for $T$ steps, i.e., sim agent models must adhere to a 10Hz resampling procedure, re-observing the updated scene and consuming their previous outputs.

2. $q^{\mathrm{world}}$ must factorize according to Eq. 1:

$$q^{\mathrm{world}}(o_t|o_{<t}^c) = \pi(o_t^{\mathrm{AV}}|o_{<t}^c)q(o_t^{\mathrm{env}}|o_{<t}^c), \tag{1}$$

where $q(o_t^{\mathrm{env}}|o_{<t}^c)$ is a traffic simulator, and $\pi(o_t^{\mathrm{AV}}|o_{<t}^c)$ is an AV policy[1]. Any submission that fails to satisfy both of these properties will not be considered on WOSAC leaderboards, as determined by

---

[1]We call this a policy because it is similar to the typical formulation of a policy in a decision process over actions, although not equivalent, because it is defined over next observations rather than current actions. It can be made equivalent to a standard policy $\pi(a_{t-1}^{\mathrm{AV}}|o_{<t}^c)$ by defining the AV's action space $\mathcal{A}$ to be equivalent to its component of the observation space, and defining an action-dependent world model $q^{\mathrm{world}}(o_t|o_{<t}^c, a_{t-1}^{\mathrm{AV}}) \doteq \delta(o_t^{\mathrm{AV}} = a_{t-1}^{\mathrm{AV}})q(o_t^{\mathrm{env}}|o_{<t}^c)$, where $\delta$ denotes the Dirac delta function.

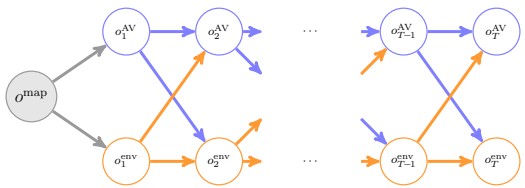

Figure 2: Graphical model of required factorization as a Bayes net: the two distributions from Eq. 1 are autoregressively interleaved: one represents the AV's "policy" $\pi(o_t^{AV}|o_{<t}^c)$, and another represents the environmental dynamics $q(o_t^{env}|o_{<t}^c)$; the graphical model represent $T$-steps of applying these two distributions. Thick outgoing arrows denote passing inputs from all parent nodes to all children.

challenge submission reports. Requiring them to be generative enables sampling from an arbitrary traffic simulator-AV policy pair. These two properties imply the probabilistic graphical model shown in Fig. 2, modified from Fig. 9 of [48]. Algorithms 1 and 2 illustrate valid and invalid submissions.

Much of the challenge of modeling $p^{world}$ lies in the fact that in many situations $s_{t-1} \in \mathcal{S}$, $p^{world}$ assigns density to multiple outcomes due to uncertainty from agents in the scene, which means that both $\pi(o_t^{AV}|o_{<t}^c)$ and $q(o_t^{env}|o_{<t}^c)$ often must contain multiple modes in order to perform well. We evaluate distribution-matching of $p^{world}$ relative to a dataset of logged outcomes. The required factorization into a AV observation-space policy and environment observation dynamics, $q^{world}(o_t|o_{<t}^c) = \pi(o_t^{AV}|o_{<t}^c)q(o_t^{env}|o_{<t}^c)$, is fairly flexible. We are agnostic to their particular structures. One noteworthy choice for the environment observation dynamics is a "multi-agent" factorization, in which $q(o_t^{env}|o_{<t}^c) = \prod_{a=1}^A \pi_a(o_t^{env,a}|o_{<t}^c)$, i.e., the environment observation dynamics factorizes into a sequence of $A$ observation-space policies, and the environment observation itself is partitioned into $A$ different components, one for each agent: $o_t^{env} = [o_t^{env,1}, \ldots, o_t^{env,A}]$.

---

**Algorithm 1** Valid: Factorized, Closed-Loop, Agent-Centric Simulation

---

**Input:** Map $o^{map}$ and traffic signals $o^{signals}$. Initial actor states $o_{-H-1:0} = \{o_{-H-1}, \cdots, o_0\}$ where each $o_t^{env} = \{o_t^{env,1}, \ldots, o_t^{env,A}\}$ for the $A$ actors in the scene.
**Output:** Simulated observations $o_{1:T} = \{o_1, o_2, \cdots, o_T\}$ for $T$ simulation timesteps.
  1: **for** $t = 1, ..., T$ **do**                              ▷ Simulate for requested number of timesteps
  2:     $o_t^{AV} \leftarrow \pi_{AV}(o_{<t}; o^{map}, o^{signals})$
  3:     **for** $a = 1, ..., A$ **do**               ▷ Produce next state for each actor at each timestep
  4:         $o_t^{env,a} \leftarrow \pi_a(o_{<t}; o^{map}, o^{signals})$
  5:     $o_t = \{o_t^{env,a} : \forall a \in 1 \ldots A\} \cup \{o_t^{AV}\}$
  6: **return** $o_{1:T} = \{o_1, o_2, \cdots, o_T\}$

---

**Algorithm 2** Invalid: Factorized, Open-Loop, Agent-Centric Simulation

---

**Input:** Map $o^{map}$ and traffic signals $o^{signals}$. Initial actor states $o_{-H-1:0} = \{o_{-H-1}, \cdots, o_0\}$ where each $o_t^{env} = \{o_t^{env,1}, \ldots, o_t^{env,A}\}$ for the $A$ actors in the scene.
**Output:** Simulated observations $o_{1:T} = \{o_1, o_2, \cdots, o_T\}$ for $T$ simulation timesteps.
  1: $o_{1:T}^{AV} \leftarrow \pi_{AV}(o_{<1}; o^{map}, o^{signals})$
  2: **for** $a = 1, ..., A$ **do**               ▷ Produce states at all future timesteps for each actor
  3:     $o_{1:T}^{env,a} \leftarrow \pi_a(o_{<1}; o^{map}, o^{signals})$
  4: **return** $o_{1:T} = \{o_{1:T}^{env,a} : \forall a \in 1 \ldots A\} \cup \{o_{1:T}^{AV}\}$

---

# 4 Benchmark Overview

## 4.1 Dataset

For WOSAC, we use the test data from the v1.2.0 release of the Waymo Open Motion Dataset (WOMD) [22]. We treat WOMD as a set $\mathcal{D}$ of scenarios where each scenario is a history-future

pair $(o_{-H-1:0}, o_{\geq 1})$. This dataset offers a large quantity of high-fidelity object behaviors and shapes produced by a state-of-the-art offboard perception system. We use WOMD's 9 second 10 Hz sequences (comprising $H = 11$ observations from 1.1 seconds of history and 80 observations from 8 seconds of future data), which contain object tracks at 10 Hz and map data for the area covered by the sequence. Across the dataset splits, there exists 486,995 scenarios in train, 44,097 in validation, and 44,920 in test. These 9.1 second windows have been sampled with varying overlap from 103,354 mined segments of 20 second duration. Up to 128 agents (one of which must represent the AV) must be simulated in each scenario for the 8 second future (comprising 80 steps of simulation).

**Agent Definition** We require simulation of all agents that have valid measurements at time $t = 0$, i.e. the last step of logged initial conditions before simulation begins. Because the test split data is sequestered, users will not have access to objects that appear after the time of handover, and so therefore could not be expected to simulate them. We require simulation of all three WOMD object types (vehicles, cyclists, and pedestrians). Objects' dimensions stay fixed as per the last step of history (while they do change in the original data).

**Submission** We do not enforce any motion model (also because we have multiple agent types), which means users need to directly report $x/y/z$ centroid coordinates and heading of the objects' boxes (which could be generated directly or through an appropriate motion model). See the Appendix for additional information on the submission format.

By allowing users to produce the simulations themselves, we reduce the burden on the user by avoiding the need to submit containerized software for an evaluation server.

## 4.2 Evaluation

Agents should generate realistic driving scenarios stochastically. We define "realistic agents" as those that match the actual distribution of scenarios observed during real-world driving. Unfortunately, we do not know the analytic form of the distribution, but we do have samples from it: the examples that make up WOMD. We therefore evaluate submissions using the approximate negative log likelihood (NLL) of real world samples under the distribution induced by the agents.

The NLL we wish to minimize is given by:

$$\text{NLL}^* = -\frac{1}{|\mathcal{D}|} \sum_{i=1}^{|\mathcal{D}|} \log q^{\text{world}}(o_{\geq 1,i} | o_{<1,i}) \tag{2}$$

However, there are two problems with trying to minimize Equation 2 exactly in our problem setting. First, $o_{\geq 1}$ is high-dimensional. Instead of trying to parameterize the entire ground truth scenario and compute its NLL under a simulated distribution, we therefore parameterize scenarios with a smaller number of component metrics (see Section 4.2.1) and aggregate them together into a composite NLL metric (see Section 4.2.2). Second, agents may support sampling but not pointwise likelihood estimation [41]. In fact, we only require challenge entrants to submit samples from their agents, and therefore have no way of knowing the exact likelihood of logged scenarios under different agent submissions. To avoid this problem, we standardize the NLL computation by fitting histograms to the 32 submitted samples of agent futures, and compute NLLs under the categorical distribution induced by normalizing the histograms.

### 4.2.1 Component Metrics

Breaking NLL$^*$ into component metrics has a few benefits. It mitigates the curse of dimensionality described in Section 4.2. It also adds more interpretability to the evaluation, allowing researchers to trade off between different types of errors.

**Time Series NLL**: Given the time series nature of simulation data, two choices emerge for how to treat samples over multiple timesteps for a given object for a given run segment: to treat them as time-independent or time-dependent samples. In the latter case, users would be expected to not only reconstruct the general behaviors present in the logged data in one rollout, but also recreate those behaviors over the exact same time intervals. To allow more flexibility in agent behavior, we use the former formulation when computing NLLs, defining each component metric $m$ as an average (in log-space) over the time-axis, masked by validity $v_t$: $m = \exp\left(-\frac{1}{\sum_t \mathbb{1}\{v_t\}} \sum_t \mathbb{1}\{v_t\} NLL_t\right)$.

| Simulation Input | Log Playback (Logged Oracle) | Wayformer (Diverse Sample) | Constant Velocity |
|---|---|---|---|

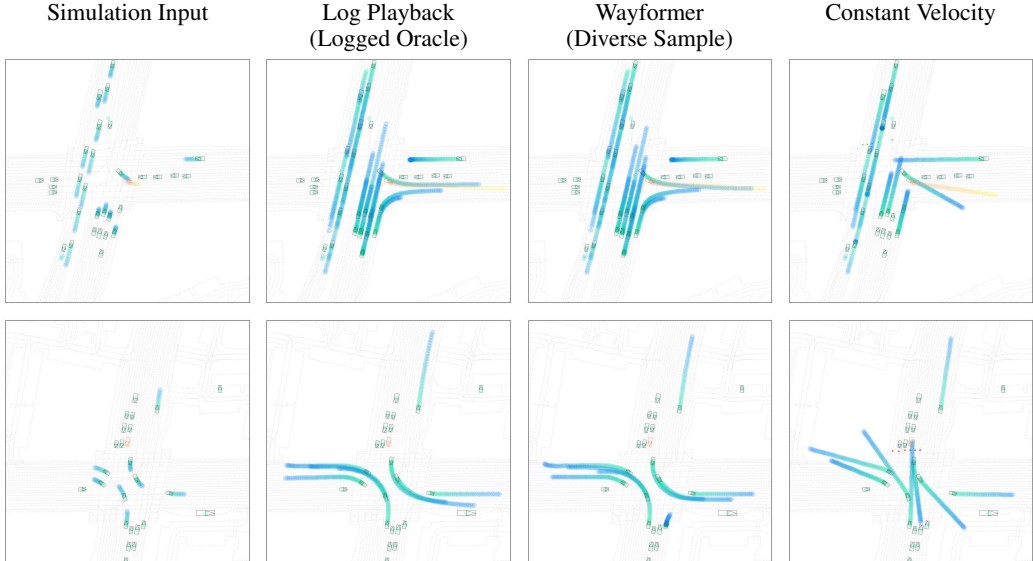

Figure 3: Visualizations of simulation results on two separate WOMD scenes (top, bottom). Results for various baseline methods are shown on WOMD's validation split, in 2d. 'Simulation input' represents the context history $o_{-H-1:0}$, whereas all other columns visualize both $(o_{-H-1:0}, o_{\geq 1})$. Two scenes are represented: one where the AV completes the execution of a left turn *(top row)* and another where the AV remains stopped at a red traffic signal *(bottom row)*. Each rendering in the second and third columns depicts the entire duration of the scene. Trajectories of environment sim agents are drawn in a green-blue gradient, and trajectories of the AV agent are drawn in a red-yellow gradient (each as a sequence of circles in a temporal color gradient).

However, we note that as a result, a logged oracle will not achieve likelihoods of 1.0, whereas in the latter formulation a logged oracle would.

**Definitions** We compute NLLs over 9 measurements: kinematic metrics (linear speed, linear acceleration, angular speed, angular acceleration magnitude), object interaction metrics (distance to nearest object, collisions, time-to-collision), and map-based metrics (distance to road edge, and road departures). Please refer to Section A.6 of the Appendix for a complete description and additional implementation details.

### 4.2.2 Composite Metric

After obtaining component metrics for each measurement, we aggregate them into a single composite metric $\mathcal{M}^K$ for evaluating submissions:

$$\mathcal{M}^K = \frac{1}{NM} \sum_{i=1}^{N} \sum_{j=1}^{M} w_j m_{i,j}^K, \qquad \sum_{j=1}^{M} w_j = 1 \qquad (3)$$

where $N$ is the number of scenarios and $M = 9$ is the number of component metrics. The component metrics $m$ and composite metric $\mathcal{M}$ are also parameterized by a number of samples $K = 32$. The value $m_{i,j}$ represents the likelihood for the $j^{th}$ metric on the $i^{th}$ example. The metric $\mathcal{M}$ is simply a convex combination (i.e. weighted average) over the component metrics, where the weight $w_j$ for the $j^{th}$ metric is set manually. In the interest of promoting safety, the weighting for collision and road departure NLLs are set to be $2\times$ larger than the weight for the other component metrics.

## 5 Experimental Results

In Figure 4 and Table 3, we present quantitative results for a handful of methods. We describe each method in more detail in the sections below.

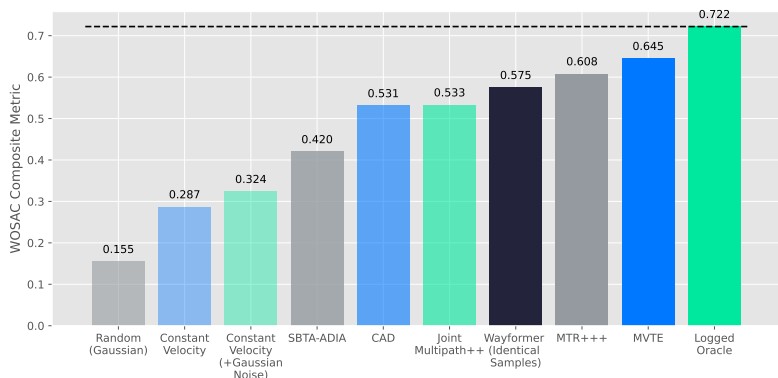

Figure 4: Challenge composite metric results of various baselines on the *test* split of WOMD.

## 5.1 Baselines

**Random Agent**: An agent that produces random trajectories $\{(x_t, y_t, \theta_t)\}_{t=1}^{T}$, for $T = 80$, with $x, y, \theta \sim \mathcal{N}(\mu, \sigma^2)$, with $\mu = 1.0$ and $\sigma = 0.1$, in the AV's coordinate frame.

**Constant Velocity Agent**: An agent that extrapolates the trajectory using the last heading and speed recorded in the provided context/history. If no two-step difference can be computed based on the valid measurements (e.g. the object appeared only at the final step of context), we set a zero speed for such agents.

**Wayformer (Identical Samples) Agent**: An agent that produces a hybrid of open-loop and closed-loop data using a Wayformer [40] motion prediction model, by executing model inference autoregressively at 2Hz. The agents execute the policy forward for 5 simulation steps, and then replan. Results with a 10 Hz replan rate instead are also shown in Table 3, and an ablation on the replan rate is provided in the Appendix. The maximum-likelihood trajectory for each agent is identically repeated 32 times to produce 32 samples. Each agent is executed by the same policy in an agent-centric frame, batched together for inference, thus complying with the required factorization.

**Wayformer (Diverse Samples) Agent**: An agent that also utilizes Wayformer [40]-generated trajectories, but samples diverse agent plans, from $K$ possible trajectories according to their likelihood, instead of selecting the maximum-likelihood choice.

**Logged Oracle**: Agent that directly copies trajectories from the WOMD test split, with 32 repetitions.

## 5.2 External Submissions

**MultiVerse Transformer for Agent simulation (MVTA)** [71]: A method inspired by MTR [55, 56] that is trained and executed in closed-loop. MVTA uses a 'receding horizon' policy with a GMM head, and consumes vector inputs.

**MVTE**: An enhanced version of MVTA [71] that samples a MVTA model from a pool of model variants to increase simulation diversity across rollouts.

**MTR+++** [44]: A hybrid open-loop/closed-loop method with a 0.5Hz replanning rate that is inspired by MTR [55, 56] and searches for the densest subgraph in a graph of non-colliding future trajectories.

For a description of other evaluated external submissions, please refer to Section A.4 of the Appendix.

## 5.3 Learnings from the 2023 Challenge

During the course of our 2023 WOSAC Challenge (March 16, 2023 to May 23, 2023), associated with the CVPR 2023 Workshop on Autonomous Driving, we received 24 test set submissions, and 16 validation set submissions, from 10 teams. We continue to receive submission queries to our evaluation server for our standing leaderboard as new teams submit new methods.

**Trends** We observed several trends among submissions. First, the challenge champion, MVTA/MVTE [71], was the only method to utilize and benefit from closed-loop training. Other methods that were trained in open-loop, such as MTR+++ [44] or our Wayformer-derived [40] baseline, found operating

Table 3: Per-component metric results on the *test* split of WOMD, representing likelihoods. Methods are ranked by composite metric on the V1 Leaderboard, rather than the previous V0 Leaderboard; Numbers within 1% of the best are in bold (excluding 'logged oracle'). * indicates a method that was received after May 23, 2023, which marked the close of the CVPR 2023 competition.

| AGENT POLICY | LINEAR SPEED (↑) | LINEAR ACCEL. (↑) | ANG. SPEED (↑) | ANG. ACCEL. (↑) | DIST. TO OBJ. (↑) | COLLISION (↑) | TTC (↑) | DIST. TO ROAD EDGE (↑) | OFFROAD (↑) | COMPOSITE METRIC (↑) | ADE (↓) | MINADE (↓) |
|---|---|---|---|---|---|---|---|---|---|---|---|---|
| RANDOM AGENT | 0.002 | 0.044 | 0.074 | 0.120 | 0.000 | 0.000 | 0.734 | 0.178 | 0.287 | 0.155 | 50.739 | 50.706 |
| CONSTANT VELOCITY | 0.074 | 0.058 | 0.019 | 0.035 | 0.208 | 0.345 | 0.737 | 0.454 | 0.455 | 0.287 | 7.923 | 7.923 |
| CONSTANT VELOCITY (+ GAUSSIAN NOISE) | 0.157 | 0.119 | 0.019 | 0.035 | 0.247 | 0.411 | 0.775 | 0.502 | 0.463 | 0.324 | 7.594 | 7.237 |
| WAYFORMER (IDENTICAL SAMPLES, 10 Hz REPLAN) [40] | 0.202 | 0.144 | 0.248 | 0.312 | 0.192 | 0.449 | 0.766 | 0.379 | 0.305 | 0.338 | 6.823 | 6.823 |
| SBTA-ADIA [38] | 0.317 | 0.174 | 0.478 | 0.463 | 0.265 | 0.337 | 0.770 | 0.557 | 0.483 | 0.420 | 4.777 | 3.611 |
| WAYFORMER (DIVERSE SAMPLES, 10 Hz REPLAN) [40] | 0.233 | 0.212 | 0.345 | 0.330 | 0.241 | 0.635 | 0.797 | 0.424 | 0.413 | 0.421 | 6.866 | 5.761 |
| CAD [14] | 0.349 | **0.253** | 0.432 | 0.310 | 0.332 | 0.568 | 0.789 | 0.637 | 0.834 | 0.531 | 3.334 | 2.308 |
| JOINT-MULTIPATH++* [70] | 0.434 | 0.230 | 0.515 | 0.452 | 0.345 | 0.567 | 0.812 | 0.639 | 0.682 | 0.533 | 5.293 | 2.049 |
| WAYFORMER (IDENTICAL SAMPLES, 2 Hz REPLAN) [40] | 0.331 | 0.098 | 0.413 | 0.406 | 0.297 | 0.870 | 0.782 | 0.592 | 0.866 | 0.575 | 2.498 | 2.498 |
| MTR+++ [44] | 0.414 | 0.107 | 0.484 | 0.436 | 0.347 | 0.861 | 0.797 | 0.654 | 0.895 | 0.608 | **2.125** | 1.679 |
| MVTA [71] | 0.439 | 0.220 | **0.533** | **0.480** | **0.374** | 0.875 | **0.829** | 0.654 | 0.893 | 0.636 | 3.925 | 1.866 |
| MVTE [71] | **0.445** | 0.222 | **0.535** | **0.481** | **0.383** | **0.893** | **0.832** | **0.664** | **0.908** | **0.645** | 3.859 | **1.674** |
| LOGGED ORACLE | 0.561 | 0.330 | 0.563 | 0.489 | 0.485 | 1.000 | 0.881 | 0.713 | 1.000 | 0.722 | 0.000 | 0.000 |

at slower replan rates necessary to obtain high composite metric results (See Table 3). Second, almost all submissions used Transformer-based methods [67], except for JointMultiPath++, which used LSTM and MCG blocks [66]. Third, all methods built primarily on top of existing motion prediction works, rather than upon existing motion planning works or sim agent methods from the literature. Only one method, MVTA/MVTE [71], incorporated aspects of an existing sim agent work, TrafficSim [60], as well as motion planning techniques, implementing a receding horizon planning policy. Thus, fourth, we observed the benefit of incorporating planning-based methods into a motion prediction framework. Fifth, most methods (excluding JointMultipath++ [70]) built upon the 2022 CVPR Waymo Open Motion Prediction challenge champion, MTR [55, 56], likely due to the open-source availability of its codebase and SOTA performance. Finally, all submissions operated in an agent-centric coordinate frame, rather than jointly sampling from a scene representation simultaneously.

**Likelihood Metrics Reward Diversity** We found that our likelihood-based metrics reward models that produce diverse futures. For example, generating 32 diverse rollouts per scene with a Wayformer model performs 11% better on our evaluation metrics than a Wayformer model that produces 32 identical rollouts per scene (see Figure 4).

**Collision Minimization as an Algorithmic Objective** Several methods designed algorithmic components to determine futures with a minimal number of collisions, e.g., MTR+++ [44] which used clique-finding in an undirected graph of collision-free future trajectories, and CAD [14], which used rejection sampling on open-loop futures that created collisions. This objective aligns with human preference, but as close calls and collisions do occur in real driving data distributions, optimizing for this objective could be seen as trimming the tail of the distribution; distracted drivers generally do exist in everyday real world driving, and in certain scenarios, one would expect a low-quality planner to perform poorly and produce collisions with sim agents, and so such should be taken into consideration for generating realistic simulations. This suggests a limitation of the WOMD [22], which has few examples from the tail distribution of real driving, and efforts to upsample collision data could prove useful. In addition, open-loop methods such as CAD [14] that prune collisions after the fact could prune collisions caused by the AV rather than by the sim agents, yielding a misleadingly optimistic view of the AV's performance.

**Composite Metric vs. (min-)ADE and ADE:** We see that among submissions to the test set, rankings by ADE and minADE and ranking by our NLL composite metric disagree. However, methods with lower minADE do tend to achieve higher composite scores; ADE does not exhibit such a trend.

**Composite Metric Results** The ordinal ranking shown in Figure 4 and Table 3 indicates that learned, stochastic sim agents outperform not only heuristic baselines but also learned, deterministic sim agents. We consider a composite metric score of 0.722 as a practical upper bound on submissions, because it involves access to test data via an oracle.

**Component Metric Results** In Table 3, we provide a breakdown of the composite metric into component metric results. As expected, the 'logged oracle' baseline achieves the highest likelihood in each of the 9 component metrics. The top performing method, MVTE [71] scored highest on all but one component metric (*linear acceleration likelihood*), where CAD [14] outperformed MVTE by 12% (likelihood of 0.253 vs. 0.222). Surprisingly, MVTE [71] has angular acceleration likelihoods within a percentage point of the 'logged oracle' (0.481 vs. 0.489). The gap between the top performing learned method (MVTE) and 'logged oracle' in both collision likelihood (0.893 vs. 1.000) and

distance-to-nearest object likelihood (0.383 vs. 0.485) indicates significant room for improvement in future work on interactive metrics.

## 5.4 Qualitative Results

In Figure 3, we provide a qualitative comparison of various baselines on two WOMD scenarios. The results indicate that the complexity of behaviors within intersections far exceeds the capability of simple heuristics to predict. Collisions are evident from the constant velocity baselines in both examples. Additional qualitative examples from other sim agent methods are shown in Section A.5 of the Appendix.

# 6 Discussion

**Limitations.** For our 2023 Challenge, we manually verified the validity of each submission according to factorization and closed-loop requirements discussed in each team's report, and we observed that the technical rules were subtle. Several of the submissions that used open-loop or hybrid open-loop/closed-loop methods may have limited applicability for some simulation applications.

Even if we had instituted a benchmark based on Docker-containerized software submissions instead of uploading output trajectory submissions, enforcing our requirements algorithmically and automatically would still be challenging. Although many properties of function calls to Dockerized software can be measured, e.g. latency, as long as any arbitrary state is maintained by the user, the system could not enforce all details of the closed-loop nature of the function call. As a result, user-submitted simulation agent software would have to adhere to strict stateless input and output data APIs. The ability to do so would assist in removing ambiguity regarding whether methods that prune collisions post-hoc qualify as closed-loop.

If a user provides containerized simulator submissions, one approach to encourage adherence to our requirements and to further incentivize closed-loop behavior would be to provide and interact with an AV policy that the user does not control. In our benchmark, the user was allowed to control the AV, albeit through an independent policy; the ability to evaluate simulator submissions on separate, held-out AV motion planning policies and on new scenarios would allow further valuable analysis.

**Future Work** Object insertion and deletion are important aspects of the simulation problem, yet we intentionally introduced an assumption of no object insertion or deletion in order to reduce the complexity of the first iteration of the WOSAC challenge for users. Motion planners trained or evaluated in a simulator must have the capability to exercise caution regarding areas of occlusion from which new objects may emerge at any timestep. In a future iteration of the challenge, we plan to introduce realism metrics that reward properly-modeled object insertion and deletion, e.g. distributional metrics on the number of vehicles appearing or disappearing at each frame, or the distance of simulated objects from the autonomous vehicle. The data distribution in the WOMD dataset already includes such object insertion and deletion.

Furthermore, we intentionally introduced an assumption of time-invariant object dimensions in our first iteration of WOSAC to simplify the modeling challenge for users. Time-variant object dimensions can be considered as a type of vehicle attribute, and object dimensions do actually change in the underlying data distribution provided in the WOMD dataset. We hope to include time-variant object dimension prediction as an aspect of the benchmark in future iterations.

As discussed in Section 5.3, given the prevalence of collision minimization algorithmic components among submissions, one may presume that collisions are not heavily represented in WOMD [22] data, or our metrics are limited in some way. Another approach would be to "fatten the tails" of the evaluation data distribution by generating synthetic, challenging initial conditions [3, 21, 23, 47, 61, 69], or mining more close calls and collisions from real driving data.

**Conclusion.** In this work, we have introduced a new challenge for evaluation of simulation agents, explaining the rationale for the different criteria we require. We invite the research community to continue to participate.

## Acknowledgments and Disclosure of Funding

No third-party funding received in direct support of this work. We thank Ben Sapp for his helpful feedback in preparing the challenge. We would like thank Mustafa Mustafa, Kratarth Goel, Rami Al-Rfou for offering consultation, models and infrastructure that accelerated our work. We thank Alexander Gorban for his assistance in developing and maintaining the evaluation server. All the authors are employees of Waymo LLC.

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

# A Appendix

In this Appendix, we provide ablations investigating the impact of replan rate of Wayformer-derived [40] sim agent baselines on the test split, as well as additional learnings from the 2023 CVPR competition. We also include additional descriptions of methods from external challenge submissions, corresponding qualitative results for such methods, and implementation details of each of the 9 component metrics we use. Finally, we describe the exact details of the dataset splits we use and give leaderboard submission instructions.

**Benchmark Versioning** In December 2023, we improved the accuracy of the collision and offroad likelihood calculation, which improved most collision likelihood scores, offroad likelihood scores, and composite metric results. This paper describes the updated scores (the V1 version of the benchmark), rather than the previous V0 scores presented at the Workshop on Autonomous Driving at CVPR 2023. Both versions of the leaderboards are available online (V1 Leaderboard, V0 Leaderboard).

## A.1 Replanning Rate Ablation Results

As shown in Table 3 of the main paper and as discussed in Section 5, we perform an ablation of the impact of replanning rate on composite metric performance for open-loop trained models on the WOMD test set. We show that a more frequent replan rate negatively impacts Wayformer-based [40] agent, irregardless of whether multiple diverse rollouts per scene are sampled or if 32 identical rollouts per scene are produced. For the identical-sample producing agent, we see a relative performance drop of 41.2% (composite scores of 0.575 vs. 0.338) when transitioning from replanning at 2Hz to 10Hz.

In Figure 5, we visualize several additional data points from a replanning interval of 100 ms to 1100 ms for the same identical-sample producing agent.

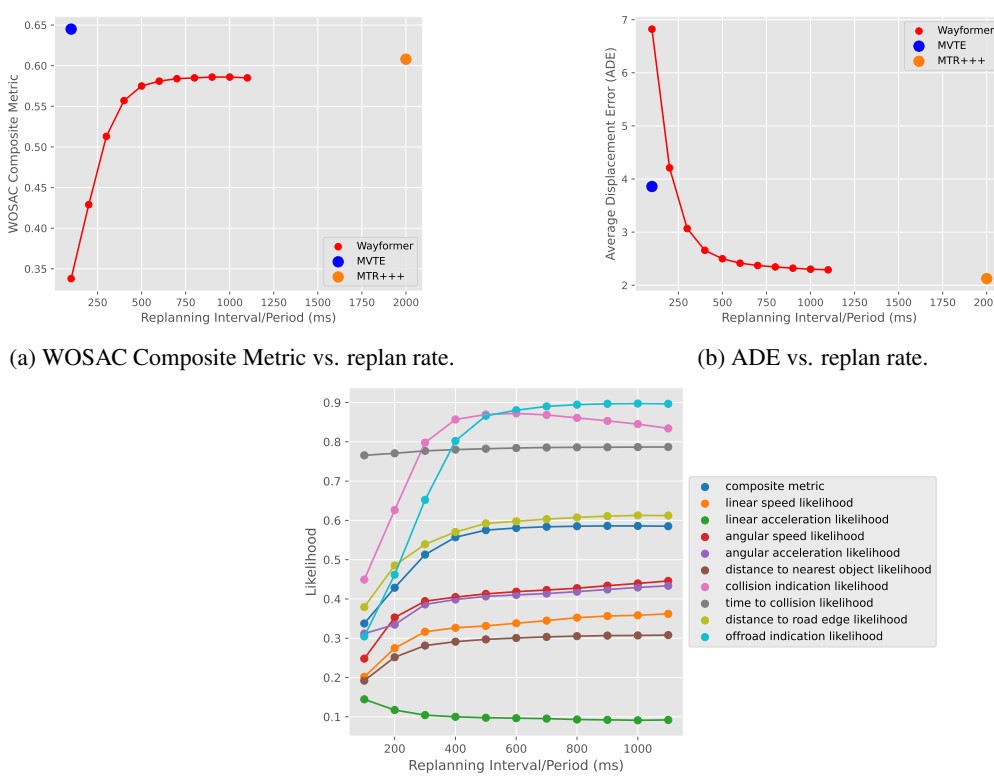

(a) WOSAC Composite Metric vs. replan rate.   (b) ADE vs. replan rate.

(c) Component likelihood metrics vs. replan rate.

Figure 5: Results at various replanning rates on the WOSAC test set for a Wayformer baseline (producing identical samples for the 32 rollouts). As the replanning rate increases from 1 Hz, then to 2 Hz, and then to 10 Hz, we observe a smooth degradation in performance.

## A.2 Additional Learnings from the 2023 CVPR Competition

**Generative Modeling** While many families of generative models exist, most challenge participants restricted their modeling to a narrow class of such models, namely GMMs. To our knowledge, for the 2023 CVPR challenge, no user submitted sim agent behavior generated by normalizing flow, GAN, or variational autoencoder (VAEs) models, or by denoising diffusion models, although such diffusion techniques have recently become more popular in the simulation agent literature [80, 81]. We expect this may change in the future as more entrants participate in the WOSAC challenge.

**Further Quantitative Analysis** We note that the top-four performing methods were executed fully in closed-loop (MVTE [71], MVTA) or in a hybrid fashion (MTR+++ [44], Wayformer [40]), outperforming the two open-loop submissions, JointMultiPath++ [70] and CAD [14], as shown in Table 4. This gap is especially clear between non-open loop methods and open-loop methods in the collision likelihood metric.

Among all baselines we evaluated, the constant velocity baseline is weakest when it comes to angular-based likelihoods. For example, it achieves close to zero likelihood on both angular speed (0.02) and angular acceleration (0.04), as opposed to the MVTE method, which achieves 0.54 and 0.38 likelihoods on the same two component metrics, respectively. This result is intuitive, as our constant velocity model does not account for any yaw rate.

## A.3 Additional Comparisons with Other Benchmarks for Autonomous Driving Behavior

In Table 4, we compare our WOSAC benchmark with other benchmarks used for evaluation of behavior models for autonomous driving.

| BENCHMARK NAME | TASK |
|---|---|
| Argoverse [10] | Trajectory Forecasting |
| INTERPRET (INTERACTION) [78] | Trajectory Forecasting |
| Argoverse2 [72] | Trajectory Forecasting |
| nuScenes [6] | Trajectory Forecasting |
| WOMD [22] | Trajectory Forecasting |
| CARLA [19] | Motion Planning |
| nuPlan [7] | Motion Planning |
| WOSAC (Ours) | Multi-Agent Simulation |

Table 4: Existing benchmarks for evaluation of behavior models for autonomous driving.

## A.4 Additional Information about Methods from External Challenge Submissions

**MultiVerse Transformer for Agent simulation (MVTA)** [71]: A method inspired by TrafficSim [60] that is trained *and* executed in closed-loop and adheres to WOSAC's factorization and autoregressive requirements. It uses a 'receding horizon' policy (i.e. predicting 1 sec. of future motion but using only the next 100 ms). Inspired by MTR [55, 56], MVTA places a Gaussian Mixture Model (GMM) head on top of a transformer-based encoder and decoder (employing the same encoder/decoder layers as implemented in MTR), consuming vector inputs. Rather than utilizing a fixed-length history as context, MVTA uses a variable-length history to potentially use all of the past data. The input agent encoding contains agent history motion state (i.e., position, object size, heading angle, and velocity) and a one-hot category mask of each agent. The prediction heads include a regression head that outputs 5 GMM parameters $(\mu_x, \mu_y, \sigma_x, \sigma_y, \rho)$, along with the velocity $(v_x, v_y)$ and heading $(\sin(\theta), \cos(\theta))$ predictions for a timestep, and a classification head that outputs probability $p$. Both heads take the query content features ($num\_query \times$hidden feature dimension) as input.

**MVTE**: An enhanced version of MVTA [71] wherein 3 variants of MVTA are trained and randomly selected to generate each of the 32 simulations, increasing simulation diversity across rollouts.

**MTR+++** [44]: A hybrid method with a 0.5Hz replanning rate and a 2 second prediction horizon. We note that MTR+++ does not fully adhere to WOSAC's closed-loop requirement, as it does not replan at a 10 Hz rate. MTR+++ also does not adhere to the policy factorization requirements, as world vs. AV policies are not separated. Inspired by MTR [55, 56], the method addresses two key limitations of MTR: inaccurate heading predictions and excessive collisions incurred by marginal predictions alone. To overcome the first issue, the authors estimate headings from $x/y$ trajectories. Second, in order to minimize collisions, the authors consider $K = 6$ trajectories predicted per agent

by MTR, and prune the exponential number of futures in a greedy fashion. As brute-force exhaustive search over the $6^N$ combinations is computationally infeasible, MTR+++ searches for the densest subgraph in a graph of non-colliding future trajectories.

First, a $6N$ by $6N$ distance matrix $D$ is constructed, where entry $D_{6m+i-1,6n+j-1}$ indicates the minimum L2 distance between the $i$th-highest trajectories of agent $m$ and the $j$th-highest one of agent $n$. Second the distance matrix is binarized by evaluating which distances correspond to collisions according to object extents. Finally, a clique-finding heuristic method finds a dense subgraph of size $N$. An ensemble of 32 fine-tuned MTR models is employed to create the 32 rollouts, each model producing a single rollout.

**Collision Avoidance Detour (CAD)** [14]: An open-loop method that builds upon an existing motion forecasting method, MTR [55, 56] to produce marginal trajectory predictions, and resamples the entire future if future agent collisions are anticipated, until a maximum number of trials is exhausted. While CAD adheres to the factorization requirement, it does not adhere to WOSAC's closed-loop requirement. Factorization of world vs. AV policies is accomplished by using different checkpoints of an MTR motion prediction model for the two agent groups, and motion of non-evaluated agents is simulated using a constant velocity model.

**Joint-Multipath++** [70]: An open-loop, scene-centric method that builds off of MultiPath++ [66], producing in a single model pass 32 rollouts, each representing an entire length-80 trajectory. JointMultiPath++ does not factorize AV vs. world policies, and thus does not fully adhere to WOSAC"s policy factorization requirements. Agent history information (positions and headings of all agents) are transformed into the AV's coordinate frame, while closest lane information is selected for each agent. In its encoder, JointMultiPath++ concatenates the output of 2 LSTMs and one MultiContextGating (MCG) [66] block to form per-agent embeddings; one LSTM is used to encode agent history, another LSTM is used to encode per-step differences in agent history, and an MCG block is used to encode agent history with corresponding timesteps. Subsequent MCG blocks fuse per-agent information with road network (polyline) embeddings. In its decoder, a series of MLP blocks transform $n$ per-agent embeddings to rollouts represented as $n \times 32 \times 80 \times 3$ output tensors. [2]

**SBTA-ADIA** Mo et al. [38]: Builds upon an existing motion prediction method [39]. This hierarchical method splits the problem into a first phase of multi-agent goal prediction, based on a GNN scene encoder, and a simple planning policy which tries to accomplish the goal closed-loop.

### A.5   Additional Qualitative Results from External Challenge Submissions

In Figure 6 and 7, we provide additional qualitative examples of simulation results from external challenge submissions.

| Simulation Input | Log Playback (Logged Oracle) | MTR+++ [44] | MVTE [71] |
|---|---|---|---|

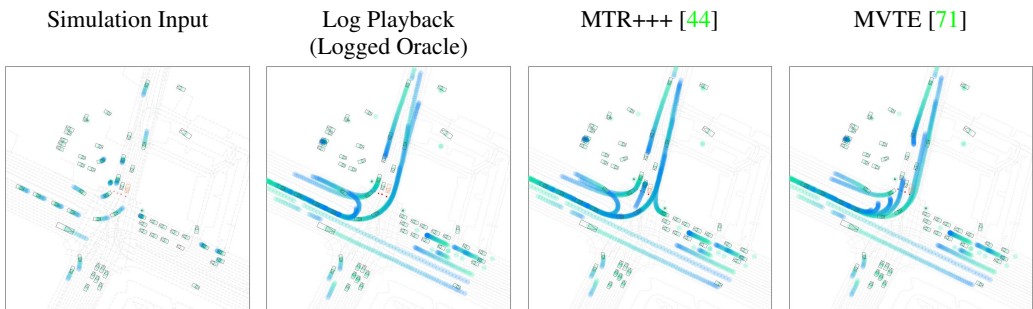

Figure 6: Two-dimensional visualization of simulation results on WOMD's test split. MVTE exhibits a collision and MTR+++ produces a near-miss. 'Simulation input' represents the context history $o_{-H-1:0}$, whereas all other columns visualize both $(o_{-H-1:0}, o_{\geq 1})$. One possible future for a single scene is represented, selected from 32 submitted rollouts, where the AV remains stopped at a red traffic signal. Each rendering in columns 2,3 and 4 depicts the entire duration of the scene. Trajectories of environment sim agents are drawn in a green-blue gradient (each as a sequence of circles in a temporal color gradient). The AV agent is drawn in orange.

---

[2]Code available at https://github.com/wangwenxi-handsome/Joint-Multipathpp.

| Simulation Input | Log Playback (Logged Oracle) | MTR+++ [44] | MVTE [71] |
|---|---|---|---|

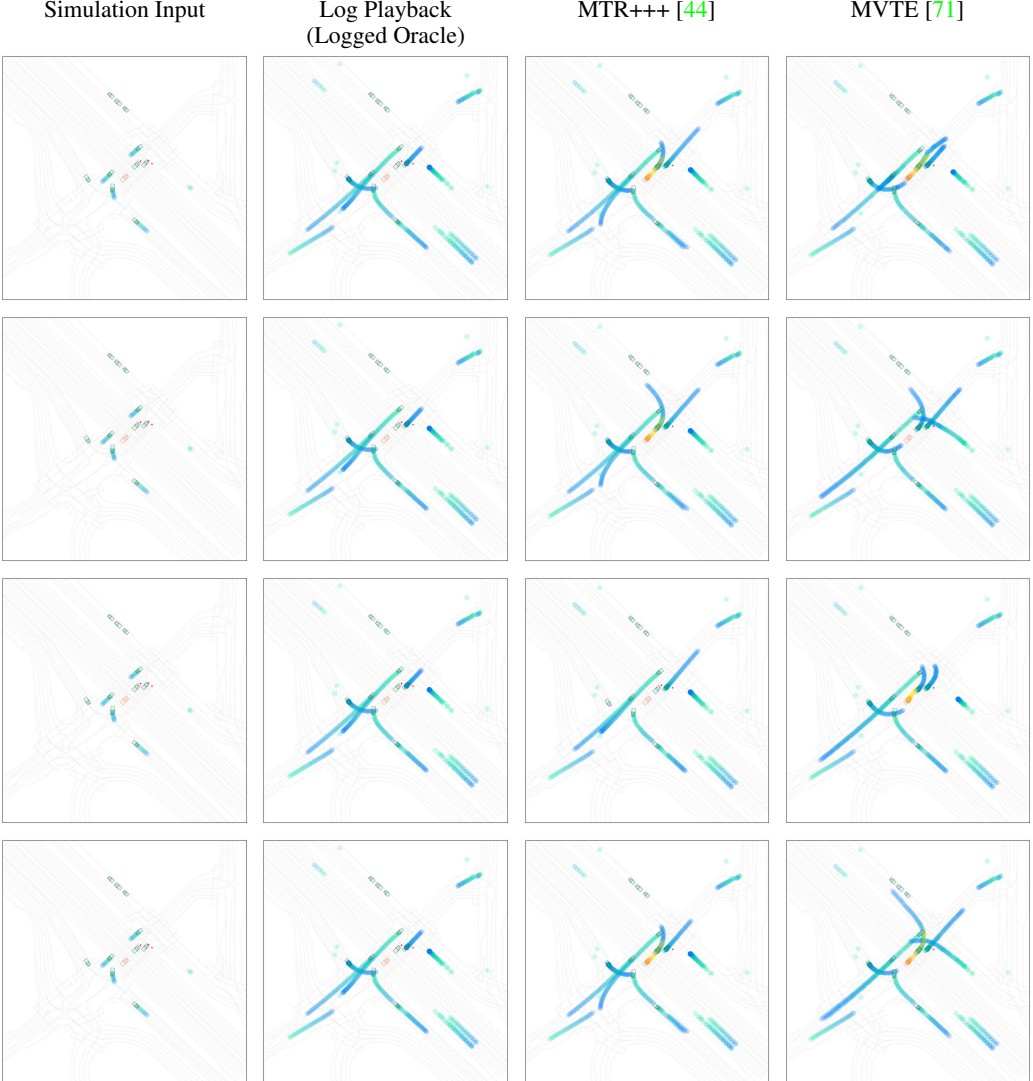

Figure 7: Two-dimensional visualization of simulation results on a single scene from WOMD's test split using various baseline methods. Four possible futures for this single scene are represented (one per row), selected from 32 submitted rollouts. 'Simulation input' represents the context history $o_{-H-1:0}$, whereas all other columns visualize both $(o_{-H-1:0}, o_{\geq 1})$. Each rendering in the second, third, and fourth columns depicts the entire duration of the scene. Trajectories of environment sim agents are drawn in a green-blue gradient, and trajectories of the AV agent are drawn in a red-yellow gradient (each as a sequence of circles in a temporal color gradient). The AV agent is drawn in orange.

## A.6 Component Metrics Implementation Details

1. Linear Speed: Unsigned magnitude of the first derivative $\|\mathbf{v}\| = \|\frac{\mathbf{x}_{t+1}-\mathbf{x}_t}{\Delta t}\|_2$ where $\mathbf{x}_t = [x_t, y_t, z_t]$, Linear speed in 3D computed as the 1-step difference between 3D trajectory points. We employ speed, rather than velocity, as velocity can either be defined w.r.t. the ego-agent's heading, or w.r.t. a global coordinate system, where velocity directions may be city-specific, based on orientation of roads w.r.t. North. Although this cannot capture objects moving in reverse, a rare behavior, we omit it for sake of simplicity.

2. Linear Acceleration Magnitude: Signed magnitude of second derivative, in 3D computed as the 1-step difference between speeds of objects. $\frac{\|\mathbf{v}_{t+1}\|-\|\mathbf{v}_t\|}{\Delta t}$.

3. Angular Speed: Signed first derivative $\omega = \frac{d(\theta_{t+1},\theta_t)}{\Delta t}$, computed as the 1-step difference in heading, where $d(\cdot)$ represents the minimal angular difference between two angles on the unit circle, i.e., $d(\cdot)$ is a distance metric on $SO(2)$ computed as $\min\{|\theta_{t+1}-\theta_t|, 2\pi-|\theta_{t+1}-\theta_t|\}$ with $\theta_t, \theta_{t+1} \in [0, 2\pi)$.

4. Angular Acceleration Magnitude: Second derivative, computed as the 1-step difference in angular speed $\omega$, as $\frac{d(\omega_{t+1},\omega_t)}{\Delta t}$.

5. Distance to nearest object: Signed distance (in meters) to the nearest object in the scene. We use Minkowski difference of box polygons, according to a simplified version of the Gilbert–Johnson–Keerthi (GJK) distance algorithm [25].

6. Collisions: Count indicating objects that collided, at any point in time, with any other object, i.e. when the signed distances to nearest objects, as described above, achieves a negative value.

7. Time-to-collision (TTC): Time (in seconds) before the object collides with the object it is following (if one exists), assuming constant speeds. An object is defined as exhibiting object-following (tailgating) behavior based on alignment conditions derived from heading and lateral distance.

8. Distance to road edge: Signed distance (in meters) to the nearest road edge in the scene.

9. Road departures: Boolean value indicating whether the object went off the road, at any point in time [60].

To prevent undefined scores from histogram bins with zero support, we employ Laplace smoothing with a pseudocount of 0.1.

**Inserted and Deleted Object Handling** In order to prevent object insertion/deletion bias between the logged and simulated data distributions during evaluation, we discard any newly spawned objects (appearing after the history interval) in the logged test set when computing the logged data distribution. The data distribution in the WOMD dataset already includes such object insertion and deletion.

### A.6.1 Evaluation Source Code References

In this section, we provide pointers to our specific implementations of the 9 metrics discussed in Section 4.2.1 of the main paper:

- Kinematic-based features: Linear speed, linear acceleration magnitude, angular speed, and angular acceleration magnitude (metrics 1,2,3,4): [Code]

- Interaction-based features: TTC and distance to nearest object (metrics 5, 6, 7): [Code] and modified GJK algorithm implementation [Code]

- Map-based features: Road departures and distance to road edge (metrics 8, 9): [Code]

An implementation of our time-series based NLL computation can be found here: [Code].

### A.6.2 Evaluation Code License and Dependencies

The WOMD [22] dataset itself is licensed under a non-commercial license (www.waymo.com/open/terms) and the evaluation code for our Waymo Open Sim Agents Challenge (WOSAC) is released under a BSD+limited patent license. See

Dependencies used include NumPy (`numpy`), the Waymo Open Dataset repository (`waymo-open-dataset-tf-2-11-0==1.5.2`), TensorFlow (`tensorflow`), TensorFlow Probability (`tensorflow_probability`), Matplotlib (`matplotlib`), TQDM (`tqdm`), Protocol Buffers (`google.protobuf`), and Python standard library imports (`os`, `tarfile`, `dataclasses`).

### A.7 Additional Information about WOMD Splits Used

We exclude 401 run segments from evaluation due to discrepancies in object counts across the `Scenario` proto and `tf.Example` formats, due to object count truncation arising from fixed-shape `tf.Example` tensors with exactly 128 object slots. We also exclude from evaluation 9 test run segments which are missing maps (however, maps are present for each scenario present in the the validation set).[3]

Table 5: Statistics of WOMD dataset [22] splits used.

|  | DATASET SPLIT | |
|---|---|---|
|  | VALIDATION | TESTING |
| ALL SCENARIO COUNTS | 44097 | 44920 |
| EVALUATED SCENARIO COUNTS | 43696 | 44520 |

### A.8 Submission Format

Submissions must be uploaded as serialized `SimAgentsChallengeSubmission` protocol buffer data[4] ("protos"). Each `ScenarioRollouts` proto within the submission must contain 32 8-second rollouts of simulation data from one scenario. A validation or test set submission may be submitted to the evaluation server.

We provide a Jupyter notebook tutorial with additional instructions and examples on how to generate a submission for a dataset split. We recommend storing multiple `ScenarioRollout`'s in each binary proto file (i.e. in each `SimAgentsChallengeSubmission` file) to prevent creating a tar.gz file with tens of thousands of files; use of 100 to 150 of such shards is recommended. Please refer to the tutorial notebook for the naming convention of these files. Submission data should be compressed as a single .tar.gz archive and uploaded as a single file.

---

[3]WOMD download instructions available at `https://waymo.com/intl/en_us/open/download`.

[4]`https://protobuf.dev/`

