# OpenReview forum: "The Waymo Open Sim Agents Challenge"
_NeurIPS.cc/2023/Track/Datasets_and_Benchmarks — NeurIPS 2023 Datasets and Benchmarks Spotlight_

### Official Review · Reviewer_xmgp · 2023-07-03

**Rating:** 6
**Confidence:** 4

**Strengths:**

1. There already has been one completed challenge in CVPR 2023, which shows the validity of the proposed dataset and benchmark.
2. Since Waymo is the largest open motion prediction dataset, this benchmark enjoys the benefits of large scale.

**Additional Feedback:**

No

**Clarity:**

The paper as well as the supplementary give enough information to understand the benchmark.

**Correctness:**

As stated in Strength 1, there already has been one completed challenge in CVPR 2023. Thus, I have no concern about the correctness.

**Documentation:**

The documentation on the website is good.

**Limitations:**

I have no concern regarding authors' discussions about the limitations and potential negative societal impact of their work.

**Opportunities For Improvement:**

Since the metrics are pure data driven, **it could happen that the method is penalized to generate diverse futures** instead of accurate ones. As a result, for the final goal of simulation - to better evaluate the AD system's ability under all possible situations, I think it is harmful. Most of collected data is trivial and might not be useful to test the system. It is those long-tail/rare/adversarial cases that are really valuable.

However, it is not feasible under the proposed benchmark since those methods which are able to generate these cases would be severely penalized under most data. In summary, I question the usefulness of the proposed benchmark to evaluate simulator.

[High-level concern] I see the leaderboard https://waymo.com/open/challenges/2023/sim-agents/ only got a few teams to participate (approximately 10). Any suggesttion to make the dataset more impact or useful to the community? This is a minor comment and will not affect my overal rating in big deal.

**Relation To Prior Work:**

There are several public multi-agent motion prediction benchmarks (e.g. INTERACTION and Argoverse 2) and closed-loop planning benchmark (nuPlan). More discussions and comparisons would better position this work and thus be beneficial for the motion prediction community to further explore the next step.

**Summary And Contributions:**

This work proposes a set of benchmark and metrics to evaluate joint multi-agent open-loop motion prediction from a simulation perspective. There is a related challenge and leaderboard hosted at CVPR 2023.

---

> ### Author Response · Authors · 2023-08-17
> **Response to Reviewer xmgp**
>
> We thank the reviewer for their time spent reviewing the paper and their valuable comments. We respond inline below:
>
> > Since the metrics are pure data driven, it could happen that the method is penalized to [for] generat[ing] diverse futures instead of accurate ones.
>
> We actually found the opposite in our evaluation – models that produce diverse samples perform best on our metrics. For example, generating 32 diverse rollouts per scene with a Wayformer model performs 11% better on our evaluation metrics than a Wayformer model that produces 32 identical rollouts per scene (see Table 3 of the Supplementary Material). We also see a similar trend with the winning MVTE model. We interpret this as follows: because the world is not deterministic, the optimal strategy to obtain accurate futures is to diversify (similar to portfolio selection in the stock market), as we lack prior knowledge of the test set. It may not appear that we have an explicit diversity metric because we name it differently (using the term “likelihood”), while other works such as BITS and TrafficSim include explicitly named diversity metrics.
>
> We have updated the paper to clarify this point.
>
> > Most of collected data is trivial and might not be useful to test the system. It is those long-tail/rare/adversarial cases that are really valuable.
>
> We agree that long-tail and rare behavior is important and that naively collecting test data would yield primarily trivial scenarios. However, we note that the Waymo Open Motion Dataset (WOMD), which we build on top of, is specifically mined for interesting scenarios (see Section 3.1 of the [WOMD paper](https://arxiv.org/pdf/2104.10133.pdf)), capturing lane changes, merges, unprotected turns, intersection left turns, intersection right turns, pedestrian-vehicle interactions, cyclist-vehicle interactions, interactions with close proximity, and interactions with high accelerations. Nonetheless, we acknowledge that there are certainly many types of rare and long-tail scenarios that are not found in WOMD, presenting an opportunity to further improve WOSAC in future work through additional scenario augmentation. We discuss this briefly in both the Future Work section (L335 to L339) and in Section 5.3 (L294 to L295).
>
> > I see the leaderboard ... only got a few teams to participate (approximately 10)
>
> To date, we have received 83 submissions from external users, from 17 teams. We agree that this challenge is the first step towards building a long term benchmark and we will continue to promote it. Given that this a more complicated topic than motion prediction that requires more effort to participate, and there are far fewer existing codebases in the multi-agent simulation space, we are pleasantly surprised that the participation is only slightly less than the perception and prediction challenges.
>
> > There are several public multi-agent motion prediction benchmarks (e.g. INTERACTION and Argoverse 2) and closed-loop planning benchmark (nuPlan). More discussions and comparisons would better position this work and thus be beneficial for the motion prediction community to further explore the next step.
>
> We agree. As we discuss in L40 to L50 and as shown in Table 1 of the main paper, there are multiple open-loop agent prediction, open-loop planning, and closed-loop planning challenges, but to the best of our knowledge, WOSAC is the first public, closed-loop sim agents challenge.
>
>
> In order to further contextualize our work, we have updated the paper to include additional comparisons with existing challenges. We welcome further suggestions of specific additional comparisons. We have also updated the text to include a reference to the INTERACTION paper and benchmark, as well as to nuPlan.
>
>
> > This work proposes a set of benchmark and metrics to evaluate joint multi-agent open-loop motion prediction from a simulation perspective
>
> We wish to point out that multi-agent simulation (as we define it) is in fact a closed-loop task (not open-loop), as it requires interaction with a reactive motion planner.

---

> > ### Comment · Reviewer_xmgp · 2023-08-19
> >
> > Thanks for authors detailed feedback. Most of the concerns have been well addressed.

---

> > > ### Author Response · Authors · 2023-08-23
> > > **Thanks to Reviewer xmgp**
> > >
> > > Thank you for the response and for taking the time to read through our comments. Given that most of your concerns have been addressed through these clarifications, would you kindly consider updating your score, or if not, be willing to share any additional questions or concerns you may have?
> > >
> > > Best wishes,
> > > The Authors

---

### Official Review · Reviewer_9P4x · 2023-07-15

**Rating:** 6
**Confidence:** 4
**Correctness:** Yes
**Clarity:** Yes

**Strengths:**

1. An evaluation framework for autoregressive traffic agents
2. An evaluation framework and an online leaderboard
3. Evaluations and analyses are provided

**Additional Feedback:**

See the comments above.

**Documentation:**

Yes

**Limitations:**

This is addressed

**Opportunities For Improvement:**

If one wants to add a new metric or do some correction to the benchmark, is this something easy that one can do directly?

**Relation To Prior Work:**

Yes

**Summary And Contributions:**

This paper proposes an open sim agents challenge. This is a public challenge. The goal is to stimulate the design of realistic simulators that can be used to evaluate and train a behavior model for autonomous driving.

---

> ### Author Response · Authors · 2023-08-17
> **Response to Reviewer 9P4x**
>
> We thank the reviewer for their time spent reviewing the paper and their comments. Regarding the procedure to add a new metric to the benchmark, users are welcome to provide proposals of new metrics via a Github “Issue” or “Pull Request” on our publicly available code repository, which contains the evaluation code (github.com/waymo-research/waymo-open-dataset), and we are open to considering such suggestions. New metrics suggested by external users would need to be carefully considered to ensure that they do not advantage the given contributor in any way. In addition, as we are instituting and maintaining the benchmark ourselves, the code maintenance and evaluation runtime computational costs associated with new metrics would also have to be carefully considered. However, as the code is publicly available, users may fork or clone the evaluation code and modify it themselves as they see fit for their own experiments.
>
> Regarding the procedure to perform a correction to the benchmark, we welcome suggestions by users about potential bugs or proposed improvements to existing code via a Github “Issue” or “Pull Request” submissions. We are committed to the code maintenance and have been responding to user requests, via email and Github issues, for several months. If a correction becomes necessary, changes to the publicly available code would be introduced, users would be notified, and all existing leaderboard entries would be re-evaluated and updated.

---

> > ### Comment · Reviewer_9P4x · 2023-08-21
> >
> > Thanks for the detailed response. All of my concerns have been addressed.

---

### Official Review · Reviewer_TuS8 · 2023-07-21
**A new Waymo challenge aiming at evaluating traffic simulation methods**

**Rating:** 8
**Confidence:** 5
**Correctness:** Dataset construction is sound and the…
**Clarity:** Easy to understand

**Strengths:**

1. This platform aims at facilitating the research of a new but important AD problem.
2. The system is well-developed and already attracts some external users.
3. The task definition is clear and complete
4. This challenge is released and maintained by a reliable organization leading AD research.

**Additional Feedback:**

N/A

**Documentation:**

Yes

**Ethics:**

No ethical issue.

**Limitations:**

Well discussed

**Opportunities For Improvement:**

One concern I have is about the data which is collected in an ego-centric way. This collection process may bring the occlusion problem, making the trajectories incomplete in one episode, which may influence the model training and evaluation. For example, a world model can never notice a car that is hidden or not exists in the first frame or in initial condition but is present in the remaining episode. Thus it is hard for the simulated scenarios to approximate the ground truth where traffic is actually influenced by cars appears lately. I believe that this can be solved by collecting data from a god's perspective.

In addition, another factor can also contribute to this failing to approximate the ground truth problem. A scenario can contain not only agents present in the initial frame but others joining the scene lately. These lately joined cars will also affect the cars' behaviors presented in the first frame. However, the auto-regressive models have little information about the addition of new vehicles and may fail to achieve a high composite metric.

Both problems are raised due to actually the same reason: the addition/deletion of vehicles that are not presented in the first frame $o_0$. How to mitigate the influence of these cars is an open problem. The most straightforward way that comes to my mind is to discard scenarios with new objects appearing/disapperaing and use scenarios with constant object IDs throughout the whole episode. But this may reduce the size of datasets. Hope authors can make improvements on this in the later version of WOSAC.

**Relation To Prior Work:**

It covers all important works I know.

**Summary And Contributions:**

Traffic simulation is important for autonomous driving (AD) evaluation, which is currently restricted to hand-crafted scenarios and replaying fixed trajectories recorded in the real world. A promising solution for realistic close-loop traffic simulation is to simulate the system dynamics with a data-driven world model built from recorded large-scale driving data. In this way, road participants such as pedestrians and vehicles cannot only retain realistic behaviors/motion trajectories but react to other agents including the ego-car. As the lead of AD research, Waymo releases this new challenge with clearly defined tasks and metrics for evaluating different ways to train world models. In the experiments, authors provide results of several baselines including some contributed by external submissions, which suggests that 1. the platform and the workflow are already mature and 2. AD community shows great interest in this challenge. As several previous Waymo challenges show great success and facilitate related research, I believe that this one is potentially impactful as well.

---

> ### Author Response · Authors · 2023-08-23
> **Response to Reviewer TuS8**
>
> We thank the reviewer for their time spent reviewing the paper and their valuable feedback. We appreciate the encouraging comments (“clearly defined tasks and metrics”, “The system is well-developed”, “The task definition is clear and complete”). We reply inline below:
>
> > A scenario can contain not only agents present in the initial frame but others joining the scene lately...the addition/deletion of vehicles that are not presented in the first frame. How to mitigate the influence of these cars is an open problem. The most straightforward way that comes to my mind is to discard scenarios with new objects appearing/disappearing...
>
> We agree that object insertion and deletion are important aspects of the simulation problem. For example, motion planners trained or evaluated in a simulator must have the capability to exercise caution regarding areas of occlusion from which new objects may emerge at any timestep. We intentionally introduced an assumption of no object insertion or deletion in the first iteration of the WOSAC challenge in order to reduce the complexity for users. In a future iteration of the challenge, we plan to introduce realism metrics that reward realistically-modeled object insertion and deletion, e.g. distributional metrics on the number of vehicles appearing or disappearing at each frame, or on the distance of simulated objects from the autonomous vehicle. The data distribution in the WOMD dataset already includes such object insertion and deletion. We have added a paragraph to clarify this in the "Future Work" section.
>
> We wish to note that we take two measures during evaluation to prevent object insertion/deletion bias between the logged and simulated data distributions. First, we discard any newly spawned objects (appearing after the history interval) in the logged test set when computing the logged data distribution. Second, we do not evaluate a simulated object after its logged counterpart undergoes a deletion stage. We have also added a paragraph to explain this in more detail.
>
> > One concern I have is about the data which is collected in an ego-centric way. This collection process may bring the occlusion problem, making the trajectories incomplete in one episode, which may influence the model training and evaluation. For example, a world model can never notice a car that is hidden or not exists in the first frame or in initial condition but is present in the remaining episode.
>
> We agree, this is an important observation. However, we note that any sensor will have a limited field of view. While an omniscient view would be possible with synthetic data, we desired to use a realistic data distribution in order to be able to the model the complexity of real-world behaviors. An overhead sensor, such as those traffic cameras and drone cameras used in the Interaction Dataset, may suffer from less vehicle-induced occlusion, but still have a limited spatial field of view.
>
> Because an autonomous vehicle will always collect data in an ego-centric field of view, a requirement of a motion planner must be the capability to reason about data collected from such a perspective, including the associated occlusion and noise. As mentioned above, naively training or evaluating a motion planner using an omniscient world model (i.e. an omniscient simulator) would introduce certain limitations, e.g. motion planners trained or evaluated in a simulator would likely lack the capability to exercise caution regarding areas of occlusion from which new objects may emerge at any timestep.

---

### Official Review · Reviewer_ZL1T · 2023-07-21
**Sim Agents Dataset Review**

**Rating:** 10
**Confidence:** 4
**Correctness:** The dataset is solid and the evaluati…

**Strengths:**

"Pose divergence" or "Simulation shift" between the AV natural driving logs and its behavior during simulation can be represented through different position, heading, speed, acceleration and etc. The paper evaluate the development of simulation agents to match the full distribution of human behavior. There is no existing benchmark for evaluation the realism of AV's simulation agents. The test data contains a large quantity of high-fidelity object behaviors and shapes produced by the state-of-the-art perception system.

**Additional Feedback:**

NA

**Clarity:**

It is better to add more discussions and references for the evaluation of generative models in the Related Work section at page 3 line 98.

**Documentation:**

There is sufficient detail on data collection and organization in the supplementary.

**Ethics:**

Reviewer do not suspect any ethical concerns with the submission.

**Limitations:**

As stated in Section 4.1, the dimensions of objects in the simulator remain fixed based on the last step of their history, while in the original data, these dimensions may vary. Therefore, authors are kindly urged to address the associated caveats and discuss any differences that arise between the simulator and the original data regarding object dimensions.

**Opportunities For Improvement:**

The WOMD consists of a collection of scenarios, with each scenario represented as a history-feature pair. To enhance clarity, the author can provide additional details on whether certain scenarios hold safety-critical implications. Furthermore, it would be valuable to explore how different scenarios might impact the distribution matching between logs and simulation agents.

**Relation To Prior Work:**

There is no existing benchmark for evaluation of sim agents. The paper is the pioneering in this field.

**Summary And Contributions:**

The WOSAC is to stimulate the design of realistic simulators that can be used to evaluate and train a behavior model for autonomous driving. The paper focus on simulating agent behavior as captured by the outputs of a perception system, such as object trajectories. The paper provide an evaluation framework for autoregressive traffic agents as well as an analysis of various baseline methods.

---

> ### Author Response · Authors · 2023-08-24
> **Response to Reviewer ZL1T**
>
> We thank the reviewer for their time spent reviewing the paper and their valuable comments. We appreciate the encouraging comments (“seminal paper”, “the paper is pioneering in this field”). We respond inline below:
>
> > As stated in Section 4.1, the dimensions of objects in the simulator remain fixed based on the last step of their history, while in the original data, these dimensions may vary. Therefore, authors are kindly urged to address the associated caveats and discuss any differences that arise between the simulator and the original data regarding object dimensions
>
> We agree. We intentionally introduced an assumption of time-invariant object dimensions in our first iteration of WOSAC to simplify the modeling challenge for users (see L49-L50 of the Introduction, where we consider time-variant object dimensions as a type of vehicle attribute). We have updated the Limitations section to reflect that object dimensions do actually change in the underlying data distribution (i.e. in the WOMD dataset). We hope to include time-variant object dimension prediction as an aspect of the benchmark in future iterations. We note that in order to prevent bias between the simulated and logged distributions, we hold each object's shape fixed across timesteps when computing features of the logged distributions. We have added a paragraph to the paper text to clarify this point.
>
> > To enhance clarity, the author can provide additional details on whether certain scenarios hold safety-critical implications. Furthermore, it would be valuable to explore how different scenarios might impact the distribution matching between logs and simulation agents.
>
> We believe each of the included scenarios could have safety-critical implications. In order to provide further insights, we will include visualizations of scenarios that ranked highest and lowest according to collision likelihood and offroad likelihood for different submissions.
>
> > It is better to add more discussions and references for the evaluation of generative models in the Related Work section at page 3 line 98.
>
> We thank the reviewer for the comment, and have updated the paper text to include additional references relevant for the evaluation of generative models.

---

### Official Review · Reviewer_58Fn · 2023-07-23
**The Waymo Open Sim Agents Challenge**

**Rating:** 8
**Confidence:** 4
**Correctness:** The claims in the paper are entirely …
**Clarity:** The paper is very well written and st…

**Strengths:**

This paper introduces the Waymo Open Sim Agents Challenge (WOSAC) for autonomous driving.

The paper notes that this is the first public competition in this important area.

Overall, I think this will be an important data set for autonomous driving.

**Additional Feedback:**

This is a strong paper in an area of need.

**Documentation:**

This paper is well written and structured. Detail on data collection, organisation, etc. is clearly provided.

**Ethics:**

I don't forsee ethical concerns at this stage. The data set is concerned with autonomous driving where ethical concerns will be an issue. That said, this paper and dataset are not yet at this stage of concern. That said, the authors would be well advised to start developing plans for addressing any potential ethical concerns with their approach.

**Limitations:**

The Discussion provided clearly outlines the limitations of this approach.

That said, given that this is the first iteration of the WOSAC competition, there is reasonable scope for further improvement.

**Opportunities For Improvement:**

The paper is somewhat preliminary and perhaps that is due to the Waymo Open Sim Agents Challenge (WOSAC) being run in May this year.

While the paper is well written, it could be improved by a more detailed list of learnings from the WOSAC 2023 competition.

**Relation To Prior Work:**

There is a good review of prior literature.

**Summary And Contributions:**

Overall, an interesting an potentially very valuable contribution to data sets for autonomous driving.

---

> ### Author Response · Authors · 2023-08-16
> **Response to reviewer 58Fn**
>
> We thank the reviewer for their time spent reviewing the paper and their valuable comments. We appreciate the encouraging comments (“This is a strong paper in an area of need”, “well-written”, “an interesting and potentially very valuable contribution”).
>
> Regarding learnings from the WOSAC 2023 competition, we thank the reviewer for the comment. We provide discussion of this in Section 5.3 (L273 to L312) and in the Limitations and Future Work (L320 to L339) sections.
>
> In order to provide additional detail, we have now added another paragraph with a more detailed list of learnings and additional discussion of the implications of the results.

---

### Decision · Program_Chairs · 2023-09-22

**Decision:**

Accept (Spotlight)

**Comment:**

Five experts reviewed this paper with all accepted recommendations. The area chairs agree that this work makes a very important contribution by introducing a new traffic simulation for autonomous driving evaluation. The reviewers did raise some valuable concerns that should be addressed in the final camera-ready version of the paper.